# Morphometric and Enzymatic Changes in Gills of Rainbow Trout After Exposure to Suboptimal Low Temperature

**DOI:** 10.3390/cimb47060457

**Published:** 2025-06-13

**Authors:** Elias Lahnsteiner, Nooshin Zamannejad, Anna Dünser, Franz Lahnsteiner

**Affiliations:** 1Federal Agency for Water Management, Institute for Water Ecology, Fisheries and Lake Research, Scharfling 18, 5310 Mondsee, Austria; nooshin.zamannejad@baw.at (N.Z.); anna.duenser@baw.at (A.D.); franz.lahnsteiner@baw.at (F.L.); 2Federal Agency for Water Management, Fishfarm Kreuzstein, Oberburgau 28, 4866 Unterach, Austria

**Keywords:** thermal stress, acclimation, morphology, enzymes, gene expression

## Abstract

The present study investigated the influence of a 30 day exposure of rainbow trout (*Oncorhynchus mykiss*) to a suboptimal low temperature of 1.8 ± 1.0 °C on their different gill characteristics (morphometry, enzyme activities, and expression of genes) in comparison to fish acclimated to 9.4 ± 0.1 °C. Morphometric analysis revealed a significant decrease in the distance between the secondary lamellae at the low temperature, which can be interpreted as a decrease in the effective gill surface. The epithelial thickness increased at the lower temperatures, which is considered a mechanism to reduce ion fluxes and save the energy costs for osmoregulation. The length of the primary lamellae, distance between the primary lamellae, length of the secondary lamellae, as well as the number of mucus cells, chloride cells, and capillaries per mm of the secondary lamella were similar between the temperature regimes. The enzymatic activities of pyruvate kinase and malate dehydrogenase were significantly increased in cold-exposed fish, whereas lactate dehydrogenase activity was higher in controls, indicating increased energy expenditure and adjustments in energy metabolism. The activities of carbonic anhydrase, caspase, Na^+^/K^+^ ATPase, and H^+^ ATPase, and the gene expressions of *hif1a*, *ca2*, *rhCG*, *slc26a6*, and *slc9a1* showed no statistically significant differences between the two temperature regimes. Therefore, it can be concluded that ammonia transport, acid–base regulation, and osmoregulation were not affected by the tested low temperature regime. These findings highlight that exposure to suboptimal temperatures induces structural and metabolic modifications in rainbow trout gills, potentially as an adaptive response to thermal stress. This study contributes to the understanding of fish acclimation to cold environments, with implications for aquaculture and ecological resilience in changing climates.

## 1. Introduction

Teleost fish studies on adaption mechanisms to suboptimal high temperatures are a focus of interest due to the worldwide increase in water temperatures associated with climate change [1,2,3]. On the other hand, suboptimal low temperatures pose significant stress challenges for fish too. However, relatively little is known about cold tolerance and potential adaptation processes. This knowledge could expand our understanding of how teleost fish cope with temperature extremes and maintain homeostasis.

The gill of teleost fish is the most important organ for gas exchange, osmoregulation, acid–base regulation, and the excretion of nitrogenous waste, and is therefore essential for their survival and adaptability [4,5]. Temperature-mediated gill remodeling has been described in several fish species such as the fresh water eel (*Anguilla anguilla*) [6], the crucian carp (*Carassius carassius*), the goldfish (*Carassius auratus*) [7], the Hoven’s carp (*Leptobarbus hoevenii*) [8], the grouper (*Epinephelus*) hybrid [9], the tilapia (*Oreochromis niloticus*) [10], and the common galaxias (*Galaxias maculatus*) [11]. A recent study demonstrated this process in relation to rainbow trout (*O. mykiss*) gill epithelial thickness, and demonstrated the activities of enzymes involved in osmoregulation, gaseous exchange, and energy metabolism change in response to the elevated temperature of 20 °C [12]. As a logical consequence of the previous cited study on *O. mykiss*, the question arises of whether low temperatures also lead to specific morphometric and metabolic changes in the gills in this species. Morphometric analysis can provide evidence of the structural adaptations of the gills to cold conditions, such as changes in the thickness of the lamellae, the surface structure or the blood supply [13]. At the same time, low temperature might affect metabolic pathways related to osmoregulation, acid–base regulation, and ammonia excretion. At low water temperatures, *C. carassius* (at 7.5 °C) and *C. auratus* (at 10 °C) gills produce an interlamellar cell mass that fills the space between the adjacent gill lamellae [7]. In *Epinephelus* hybrid (exposed to 22 °C) the gill secondary lamella thickness [9] and in *A. anguilla* (exposed to 5 °C) the branchial water–blood barrier thickness [6] increase in the lower suboptimal temperature range.

The present study investigated if and how a 30 day exposure to 1.8 ± 1.0 °C affected the gills of rainbow trout. The primary objective was to identify any pathological changes in the gills triggered by the suboptimal low temperature as well as to detect potential morphometric or cellular alterations within the gills. In addition, the study measured the key enzymes for various metabolic processes. These included the enzymes indicative of ion transport (Na^+^/K^+^ ATPase, H^+^ ATPase), acid–base homeostasis, and pH regulation (carbonic anhydrase), glycolysis (pyruvate kinase, lactate dehydrogenase), the citric acid cycle (malate dehydrogenase), and apoptosis (caspase 3), as well as the expression of the gene carbonic anhydrase II (*ca2*), of the rhesus blood group C glycoprotein (*rhCG*), of the solute carrier families 9 and 26 (*slc9a1*, *slc26a6*), and of the hypoxia-inducible factor 1-α (*hif1a*). The described genes were selected as they gave deeper insight in the adaption processes of ion (*slc9a1*, *slc26a6*) and ammonium transport (*rhCG*), and of ion (*ca2*) and oxygen homeostasis (*hif1a*) to low temperatures.

## 2. Materials and Methods

### 2.1. Fish

The experiments were carried out in the fish farm Kreuzstein, Oberburgau 28, 4866 Unterrach, Austria.. Twelve-month-old rainbow trout (146.18 ± 43.51 g) were used which were acclimated to 9.4 °C ± 1 °C over their whole lives and which had a natural photoperiod. The experiments were part of a previously published study characterizing hematological and physiological changes in rainbow trout at a critical low temperature [14].

### 2.2. Temperature Exposure

The procedure of exposing rainbow trout to low temperatures has been described in detail in the previous study [14]. Briefly, experiments were conducted in stream channels (190 × 25 × 35 cm, length × width × height) under flow-through conditions in January. Four stream channels were used. Two channels were supplied with groundwater of a constant temperature of 9.4 ± 0.1 °C and they served as controls. The other two stream channels were initially also supplied with groundwater. Then, the groundwater was mixed with creek water, another water source of the fish farm, which had temperatures < 3 °C during January. Groundwater was gradually cooled to the creek water temperature over a period of 10 days (minus 0.8 °C/day) by mixing the two water sources. When the water temperature had reached creek temperature (i.e., when the stream channels were supplied solely with creek water), the experiment was started. The duration of the experiment was 30 days. The water temperature of the control and experimental stream channels is shown in Figure 1. Rainbow trout were fed a commercial trout diet at a ratio of 1.0% of the body weight. Stream channels were inspected two times daily (8:00 a.m., 4:00 p.m.) and cleaned according to the hygienic concept of the fish farm. Experiments were carried out in accordance with Austrian regulations governing animal welfare and protection and with the European Union (EU) directive 2010/63/EU for animal experiments.

### 2.3. Sampling

At the end of the experiment, 10 fish per stream channel were haphazardly sampled and euthanized with 0.3% MS 222. The heart ventricle was opened to bleed the fish out as erythrocytes occurring in gill tissue might interact with enzyme activity and gene expression analysis. The second gill arch was excised as described previously [15]. For enzymatic analysis, 10 samples per treatment were processed. For morphometric and histological analyses, 7 samples per treatment were used, and for gene expression analysis, 6 samples per treatment. The reduced sample sizes for the latter two analyses were chosen to minimize costs and labor requirements.

The gill arch of the right body side was used for morphometric investigations, and the gill arch of the left body side for the analysis of enzyme activities and gene expressions. For morphometric investigations, the gill arch was fixed in 0.1 mol/L of cacodylate buffered 4% glutaraldehyde solution (pH 7.4) at 4 °C (5 samples per stream channel) and for enzymatic analysis the tissue was immersed in 100 mmol/L of Tris buffer (pH 7.4) (5 samples per stream channel) containing 5 mmol/L of EDTA and 0.5% Triton X-100, and this was also conducted for gene expression analysis in RNA later (3 samples per stream channel). Enzyme and RNA samples were stored at −25 °C.

### 2.4. Gill Histology and Morphometry

Morphometric analysis of the gills conformed to a previous study [15] to allow data comparability. It was performed on 7 individuals per treatment. First, the fixed samples were photographed in a stereomicroscope at 10–25× magnification together with a calibration slide, as shown in Figure 2a. From the digitized micrographs, the length of the primary lamellae (Figure 2a) and secondary lamellae (Figure 2b) and the distance between the primary lamellae were measured. To determine the distance between the single primary lamellae, 20 primary lamellae were counted and marked. The distance from the 1st to the 20th primary lamella was measured at the base of the gill arch (Figure 2a) and divided by 20. The secondary lamellae were measured as shown in Figure 2b. The distance between the primary lamellae was 1 measurement per individual, while all other parameters were measured 5 times per individual on different gill lamellae.

Then, the samples were processed for histological analysis. The fixative was rinsed out, and the samples were decalcified in 10% EDTA solution, dehydrated in a graded series of ethanol, and embedded in Technovit 7100. Five histological sections with a thickness of 5 µm were prepared from 3 different gill regions per individual and stained with 0.1% toluidine blue diluted in 0.1 mol/L of phosphate buffer (pH 7.2). From each gill region, 1 high-quality section was selected and photographed at 400- and 1000× magnification. Also, the histological analysis procedure conformed to the previously published study [14]. The distances between the secondary lamellae (Figure 2b) and the thickness of the epithelium were measured (Figure 2c). The number of capillaries, chloride cells, mucus cells, and capillaries in defined length sections of the secondary lamella were counted (Figure 2b) and extrapolated to 1 mm. Mucus cells and chloride cells are shown in Figure 2c,d. The measurement number was 15 per section and therefore 45 per individual. For the different analyzed parameters, the mean per individual was calculated for further statistical analysis. All measurements were performed in the Image J program (1.54g).

### 2.5. Enzymatic Analyses

Gill tissue was homogenized from 10 individuals per treatment using a Dounce-type tissue homogenizer (VWR International, Vienna, Austria) and centrifuged for 10 min at 1000× *g*, and the supernatant representing the crude enzyme extract of the sample was used for analysis. Protein concentration was determined colorimetrically [16] and the enzyme activities were determined referring to the protein concentration of the samples. The temperature of the enzymatic assays corresponded to the water temperature of the fish, i.e., 9 °C for control fish and 1.8 °C for fish exposed to decreased temperatures. Adequate blanks (no substrate, no sample, specific inhibitors defined below) were used to run all enzymatic assays. Optimal substrate and inhibitor concentrations were determined in preliminary experiments. Lactate dehydrogenase (substrate: 2 mmol/L pyruvate, pH 7.4), pyruvate kinase (substrate: 3 mmol/L phosphoenolpyruvate, assay pH: 7.8), and malate dehydrogenase (2 mmol/L malate, assay pH: 9.4) were assayed UV-spectrophotometrically [17]. ATPases were determined by NADH-coupled enzymatic assays monitoring the amount of liberated ADP [18] with 5 mmol/L of ATP as the substrate and an assay pH of 7.6. Na^+^/K^+^ ATPase activity was measured in the presence of 100 mmol/L of NaCl and 20 mmol/L of KCl with 2.5 mmol/L ouabain as the specific inhibitor [19]. H^+^-ATPase was measured as the bafilomycin-sensitive ATPase activity using 50 μmol/L of bafilomycin as the inhibitor [19]. Carbonic anhydrase was determined colorimetrically (substrate: 2 mmol/L 4-nitrophenyl acetate, assay pH: 7.5) using 2 mmol/L acetazolamide as specific inhibitor to exclude non-specific esterase activity [20]. Caspase 3 was determined with 1.5 mmol/L of the chromogenic substrate Ac-Asp-Met-Gln-Asp-pNA (Sigma-Aldrich, Vienna, Austria) at a pH of 7.4 according to the assay procedure of [21]. Absorbance was measured with a Multiskan^TM^ FC Microplate Photometer (Thermo Fisher Scientific, Waltham, MA, USA).

### 2.6. Gene Expression Analysis

Total RNA was extracted from 6 individuals per treatment using an RNA isolation Kit from Machery-Nagel (Düren, Germany). The quality of the extracted RNA was determined with a Qubit 4 Fluorometer (Thermo Fisher Scientific, Vienna, Austria). RNA samples with an integrity reading > 7 were used for analysis. RNA was reverse transcribed into cDNA using a RevertAid First Strand cDNA Synthesis Kit (Thermo Fisher Scientific) with oligo-dT and random primers. Real-time PCR was performed using a qTower3 G (Analytic Jena, Jena, Germany) and a Sybr^®^ Green Jump StartTM Taq ReadyMixTM (Sigma-Aldrich, Vienna, Austria). The mRNA expression of *hif1a* (hypoxia-inducible factor-1α), *RhCG* (rhesus glycoprotein-related genes), *slc9a1* and *Slc26a6* (solute carrier family), and *ca2* (carbonic anhydrase II) were analyzed. The 28S ribosomal protein S32 (*RibS32*) was used as an internal reference gene. Primer sets for these six genes were designed utilizing the Primer 3 tool [22] and inputting mRNA sequences in the FASTA format obtained from NCBI. The specificity of the primers was further analyzed using the Primer Blast tool from the National Library of Medicine [21] to reduce off target amplification and to ensure the optimal performance of the primer pairs (Table 1). Primer sequences are reported in Table 1. The PCR conditions were an initial denaturation step of 95 °C for 10 min followed by 40 cycles consisting of 95 °C for 20 s, 60 °C for 20 s, and 72 °C for 10 s. The Ct values were calculated using the supplied analysis software. The relative transcription levels of genes of interest were calculated by the 2^−∆∆Ct^ method [23].

### 2.7. Statistics

All data are presented as mean ± standard deviation. Morphometric and enzymatic data had a normal distribution, as revealed by the Shapiro–Wilk test. Therefore, Student’s *t*-test was used for data comparison (set from 9.4 ± 0.1 °C versus set from 1.8 ± 1.0 °C). Data were considered significantly different at a probability level of *p* ≤ 0.05. Gene expression data of the fish kept at the two temperature regimes are reported as geometric means and are compared using the Mann–Whitney U test at a probability level of *p* ≤ 0.05.

## 3. Results

### 3.1. Gill Morphometry

The width of secondary lamellae, the distance between the secondary lamellae, and the mean epithelial thickness significantly increased in rainbow trout exposed to the low temperature for 30 days compared to those kept at the acclimation temperature (Table 2, raw data see Appendix A, data set morphometry). However, parameters such as the length of the primary lamellae, distance between the primary lamellae, and length of the secondary lamellae, as well as the number of mucus cells, chloride cells, and capillaries, and the number of necrotic cells per mm of the secondary lamella were similar for the two temperature regimes.

### 3.2. Enzymatic Activities and Gene Expressions

Pyruvate kinase and malate dehydrogenase activities significantly increased in rainbow trout exposed to 1.8 °C for 30 days compared to those acclimated to 9 °C, whereas lactate dehydrogenase was significantly higher in the control group (Table 3, raw data see Appendix A, data set enzyme activities). Carbonic anhydrase, caspase, Na^+^/K^+^ ATPase, and H^+^/ATPase showed no statistically significant differences between the two treatments (Table 3, raw data see Appendix A, data set enzyme activities). Relative changes in mRNA expression of *hif1a*, *rhCG*, *slc9a1* and *slc26a6*, and *ca2* did not differ between 9.4 ± 0.1 °C and, 1.8 ± 1.0 °C (Figure 3, raw data see Appendix A, data set gene expression).

The gene expressions of *ca2*, *hif1a*, *rhCG*, *slc12a2*, and *slc9a1* were similar between 9.4 °C and 1.8 °C (Table 3).

**Figure 3 cimb-47-00457-f003:**
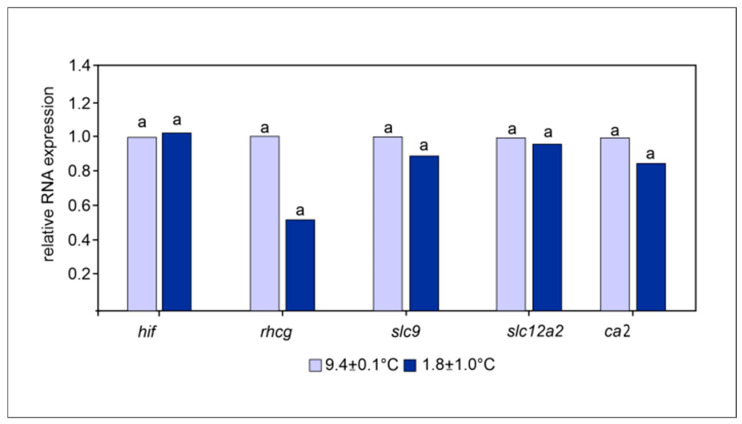
Relative changes in mRNA expression of *hif1a*, *rhCG*, *slc9a1* and *slc26a6*, and *ca2* between rainbow trout acclimated to 9.4 ± 0.1 °C and those exposed for 30 d to 1.8 ± 1.0 °C. Internal reference gene: 28S ribosomal protein *S32*. Data are presented as geometric means (n = 6); those superscripted by different letters are significantly different (*p* < 0.05).

## 4. Discussion

A previous study demonstrated that 1.8 °C was a suboptimal low temperature for rainbow trout, inducing thermal stress [14]. This was indicated by an increased mortality rate, decreased growth rate, and an upregulation of the heat shock protein *hsp90* gene in comparison to the acclimation temperature. The presented results indicate that the mentioned low temperature regime also induced changes in the gills which concerned their morphometry, metabolism, and gene expression. On the morphological level, the distance between the secondary lamellae, the width of secondary lamellae, and the mean epithelial thickness were increased. The width of the secondary lamellae and mean epithelial thickness are parameters associated with each other, as a thickened epithelium causes an increase in the width of the secondary lamellae. The changes in comparison to the 9.4 °C control ranged from 25% in the secondary lamellae width and 16% in epithelial thickness to 14% in secondary lamellae distance. As discussed in earlier studies, the gill epithelium represents a compromise between conflicting demands. Reduced epithelial height enhances gaseous exchange and water/ion fluxes but also increases osmoregulatory costs, whereas the reverse adaptations decrease gaseous exchange efficiency in favor of osmoregulatory costs [24,25]. As an increase in gill epithelium height was also observed after exposure to elevated temperature [9,12] and to environmental pollutants [26,27,28], these changes may be considered as a general reaction mechanism against suboptimal environmental conditions. The energy saved by the reduction in osmoregulation might be invested in the maintenance of cell homeostasis, becoming costlier under suboptimal conditions. On the morphometrical level, the present study also gives some indication of changes in the gill surface area. The increase in the distance between secondary lamellae implies that a lower number of secondary lamellae adhere to the primary lamellae, as the lengths of the primary lamellae are constant. This could be interpreted as a reduction in the gill surface area at low temperatures. This interpretation is still hypothetical, as the gills represent a complex three-dimensional structure difficult to capture by routine analytical methods [29]. Also, the mechanisms leading to the reduction or degeneration of secondary filaments are unknown. Generally, the reduction in gill surface could be associated with lower oxygen demand due to the decreased metabolic rate of rainbow trout at 1.8 °C [14]. Changes in the gill surface area in response to oxygen demand have also been observed in other studies on *Ctenopharyngodon idella* due to hypoxia [30], in *C. auratus* due to the sustained effect on subsequent swimming performance [31], and in *C. carassius and C. auratus* gills in response to low oxygen demand [7]. The number of chloride cells, the sites of Na^+^ and Cl^−^ transport, and the sites of acid–base regulation [32,33] were not affected by the tested temperature regime. These morphological data can be considered in agreement with enzymatic activity data on Na^+^/K^+^ ATPase and H^+^ ATPase and on gene expression data of *sclc9* and *slc26a6* (see below). Also, the number of mucus cells was similar between the tested temperature regimes. Gill mucus protects the epithelium against environmental factors and pathogens and therefore the number of mucous cells is considered an important evaluation tool for gill health [34]. Finally, the number of capillaries in the secondary lamellae also did not differ, which is an indication that vascularization remained unchanged. However, the capillary architecture of the gills is also a complex three-dimensional network [35] which cannot be fully captured by two-dimensional analytical methods.

The enzymatic activities of carbonic anhydrase, Na^+^/K^+^ ATPase, and H^+^ ATPase activity did not differ between rainbow trout exposed to 1.8 °C and those acclimated to 9.4 °C. The results of the carbonic anhydrase activity were also corroborated by gene expression data. Carbonic anhydrase catalyzes the reversible conversion of carbon dioxide to carbonic acid. In the gills of freshwater teleosts, it is essential for regulating pH by removing excess H^+^, facilitating Na^+^ uptake via H^+^-ATPase activity, maintaining Cl^−^ balance through HCO_3_^−^/Cl^−^ exchange, and supporting CO_2_ excretion for efficient respiration [36,37]. Na^+^/K^+^-ATPase and H^+^-ATPase are involved in osmoregulation and function as serial membrane ion-pumps. The H^+^-ATPase actively pumps H^+^ ions out of the cells into the surrounding water. It creates an electrochemical gradient that facilitates the passive uptake of Na^+^ ions from water via Na^+^ channels [18,38]. Na^+^/K^+^-ATPase actively pumps Na^+^ out of the cell into the blood and K^+^ into the cell. It keeps the intracellular Na^+^ concentration low, which enhances Na^+^ uptake from water through the apical Na^+^ channels [18,38]. Similarities in enzyme activities between the two temperature regimes indicate that the described processes of pH and osmoregulation are not affected by 30 days of exposure to 1.8 °C in rainbow trout.

The *slc9a1* and *slc26a6* gene families encode ion transporters that, in addition to the mentioned ATPases, also play essential roles in ion homeostasis, pH regulation, and osmoregulation. *slc9a1*, the sodium/hydrogen exchanger, is a secondary active transport system. It uses the Na^+^ gradient created by Na^+^/K^+^-ATPase and exchanges Na^+^ into the cell for H^+^ out of the cell [39]. Therefore, this exchanger works alongside H^+^-ATPase and Na^+^/K^+^-ATPase and regulates intracellular pH by removing H^+^ and contributes to maintaining sodium homeostasis [39]. *Slc26a6* exchanges Cl^−^ for HCO_3_^−^, contributing to maintaining chloride balance and pH homeostasis [40,41]. It facilitates Cl^−^ uptake from water, plays a role in acid–base regulation, and works in tandem with carbonic anhydrase [40,41]. In rainbow trout exposed to 1.8 °C, the expression of *sclc9* and *slc26a6* genes were similar to at 9 °C. These results, together with the results on enzyme activities, are indications that osmo- and pH regulation were not disturbed in rainbow trout at 1.8 °C. However, the impairment of osmoregulation at a low water temperature has been described in several studies in Salmonidae, *O. mykiss* [42], *Salmo trutta* [43], and *Salmo salar* [44].

Pyruvate kinase, a rate-controlling enzyme of glycolysis (catalyzing the transfer of a phosphate group from phosphoenolpyruvate to adenosine diphosphate), showed higher activity at 1.8 °C compared to 9.4 °C, similar to malate dehydrogenase, a key enzyme of the citric acid cycle catalyzing the oxidation of malate to oxaloacetate. These data demonstrate an increased energy demand of the gill epithelium at 1.8 °C. They can be interpreted as an indication that a temperature of 1.8 °C reflects suboptimal conditions for the gills and functionality can only be maintained by increased energy expenditure.

The *hif1a* gene encodes the transcription factors regulating genes involved in cellular adaptation to low oxygen conditions. The main regulative adaptations concern oxygen homeostasis, angiogenesis, erythropoiesis, metabolism (shifts from aerobic respiration to anaerobic glycolysis). The expression of *hif1a* did not differ between fish acclimated to 9.4 °C and fish exposed to 1.8 °C which ascertains the morphometric data that no hypoxic conditions occur due to low-temperature exposure. Further, these data also conform to the decrease in lactic acid dehydrogenase at 1.8 °C, an enzyme responsible for converting pyruvate to lactate for energy production under anaerobic conditions.

The *Rh* genes encode Rh-associated glycoproteins (RhAG, RhBG, RhCG), which are primarily involved in ammonia transport (gills, kidney, liver) and gas exchange (erythrocytes) in various tissues. In fish gills, *rh* encodes Rh proteins which play a key role in ammonia excretion in teleost fish [45]. While some ammonia passes the cell membranes by simple diffusion, its most significant portion passes through Rh protein channels, and this fraction is increased when Rh protein expression is induced by ammonia accumulation [45,46]. A previous study showed that feed uptake and growth in rainbow trout is greatly reduced at 1.8 °C. Since dietary protein is a major source of amino acids and subsequently of ammonia waste in fish [46], it would be expected that *rhcg* was downregulated in the gills of rainbow trout exposed to 1.8 °C. However, this hypothesis has to be rejected, as *rchg* did not differ between rainbow trout kept at 9.4 °C and 1.8 °C.

Finally, there were no indications that necrotic or apoptotic processes occurred in the gills of rainbow trout exposed to 1.8 °C in comparison to fish acclimated to 9.4 °C. The number of necrotic cells as well as the enzyme activities of caspase 3 did not differ between the two temperature regimes.

## 5. Conclusions

The gills of rainbow trout exposed to 1.8 °C for 30 days revealed morphometric adaptions. These concerned an increase in epithelial thickness considered to be a mechanism to reduce ion fluxes and save energy costs for osmoregulation. The increase in the distance between the secondary lamellae can be interpreted as a decrease in the effective gill surface, a potential adaption mechanism to low oxygen demands at 1.8 °C. From the enzyme activity and gene expression data, it can be concluded that ammonia transport, acid–base regulation, and osmoregulation were not affected by the tested low temperature regime. The energy expenditure to maintain gill functionality was higher at 1.8 °C than at 9.4 °C, indicated by the increase in pyruvate kinase and malate dehydrogenase activity.

Although rising temperatures due to climate change pose a well-known threat to fish in both wild populations and aquaculture, cold events also occur and should not be overlooked. The present study, along with a previous investigation, provides initial evidence that significant—and often detrimental—changes take place in fish at both the organismal and molecular levels. These changes are not negligible, as they can impact the dynamics of natural fish populations and aquaculture systems under extreme environmental conditions. Notably, as fish acclimate to higher water temperatures, their sensitivity to cold may simultaneously decrease.

## Figures and Tables

**Figure 1 cimb-47-00457-f001:**
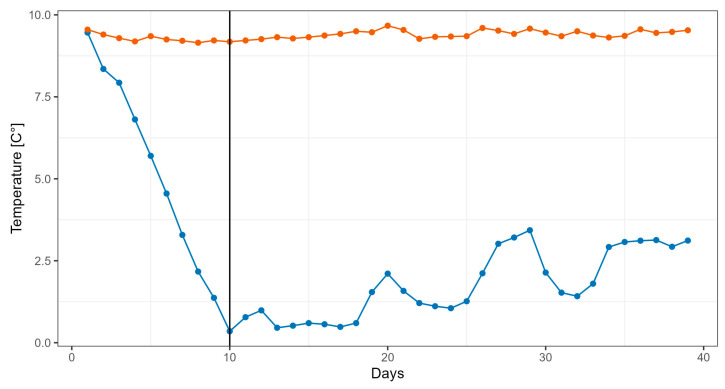
Temperature in the control (red line) and temperature exposure (blue line) stream channels. Data are the mean of the 2 replicates. Black vertical line indicates start of experiment.

**Figure 2 cimb-47-00457-f002:**
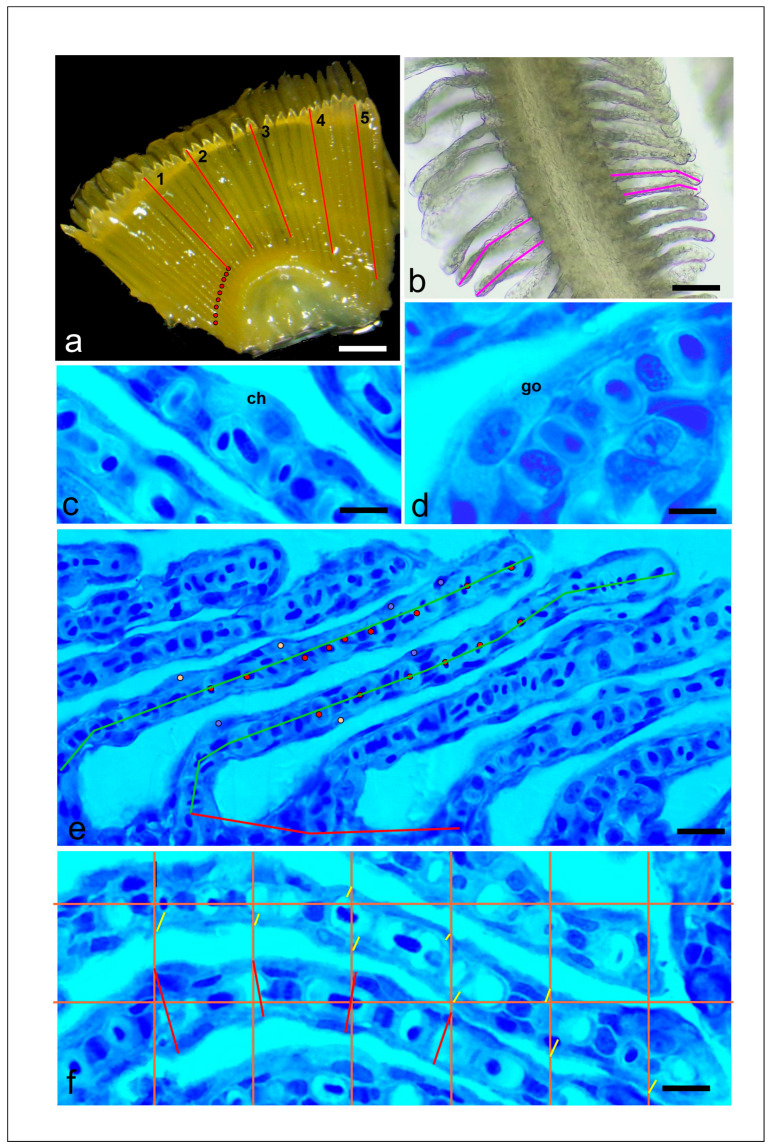
Morphometric landmarks measured in gills of rainbow trout. (**a**) Measurement of the length of the primary lamella (every 5th lamella was measured, see example 1–5) of the gill arch (red lines) and of the distance between primary lamellae. Ten primary lamellae were counted and marked (red circles). The distance from the 1st to the 10th primary lamella was measured at the base of the gill arch and divided by 10. Stereomicroscopic micrograph. Scale bar = 1500 µm. (**b**) Measurement of the length of the secondary lamellae (pink line). Stereomicroscopic micrograph. Scale bar = 150 µm. (**c**) Chloride cell (ch). Scale bar = 8 µm. Histological section. (**d**) Mucus cell (go). Scale bar = 8 µm. Histological section. (**e**) Measurement of the distance between the secondary lamellae (orange lines) and the counts of mucus (yellow circle), chloride cells (blue circle), and capillaries (red circle). The secondary lamella was measured and the cell types and capillaries were counted and their number extrapolated to 1 mm. Scale bar = 15 µm. Histological section. (**f**) Measurement of secondary lamella and epithelium thickness. A grid was placed over the section and measurements were taken where the grid lines intersected the epithelium. Epithelial thickness measurements (yellow lines) are marked for the whole picture; secondary lamella measurements (red lines) are only marked for 5 examples. Scale bar = 12 µm. Histological section.

**Table 1 cimb-47-00457-t001:** Primers used for RT–qPCR analysis. All primers used had an efficiency over 90%.

Gene	Forward and Reverse Primer (5′-3′)	Amplicon Length [bp]
*ca2* (carbonic anhydrase II)	CAAGGAGTCAATCAGCGTCA TCCGTGTTTGGATCTCTTCC	226
*hif1a* (hypoxia-inducible factor 1-α)	CAGGTCCAGACTCCTTCAGC CAACATCTCCAGGTCCAGGT	200
*ribS32* (28S ribosomal protein S32)	TCCTCGATGGTGGGGGCCTG AGACCCCCTGGCCAATCCGG	
*rhCG* (rhesus blood group C glycoprotein)	GAGCTTTTCCTGAGCATTGG GCCACAGAATGAGGATTGGT	160
*slc26a6* (solute carrier family 26 member 6)	GTGACGTCATTTCGGGAGTT GCCATGCTCAGAACAACAGA	233
*slc9a1* (solute carrier family 9 member A1)	TTCCTGGACCATCTCCTCAC AGCTCCATCTTGCGGTAGAA	170

**Table 2 cimb-47-00457-t002:** Changes in the morphometric parameters of the gill of the rainbow trout exposed to 1.8 ± 1.0 °C for 30 days in comparison to fish acclimated to 9.4 ± 0.1 °C. Data are mean ± standard deviation (n = 10). Data superscripted by different letters are significantly different (*p* < 0.05).

Morphometric Parameters	9.4 ± 0.1 °C	1.8 ± 1.0 °C
Parameters with significant differnces		
Width of secondary lamellae [µm]	14.34 ± 2.15 ^a^	18.05 ± 2.04 ^b^
Distance between secondary lamellae [µm]	27.44 ± 2.46 ^a^	31.18 ± 2.79 ^b^
Mean epithelial thickness [µm]	2.99 ± 0.65 ^a^	3.49 ± 0.82 ^b^
Parameters without differences		
Length of primary lamellae [µm]	5227.33 ± 505.50 ^a^	5269.67 ± 561.97 ^a^
Distance between primary lamellae [µm]	339.45 ± 25.73 ^a^	317.81 ± 28.43 ^a^
Length of secondary lamellae [µm]	270.45 ± 12.73 ^a^	282.44 ± 15.51 ^a^
No. of mucus cells/mm	8.97 ± 1.98 ^a^	9.67 ± 2.31 ^a^
No. of chloride cells/mm	10.50 ± 1.46 ^a^	12.12 ± 2.14 ^a^
No. of capillaries/mm	58.17 ± 3.13 ^a^	61.61 ± 4.00 ^a^
No. of necrotic cells/mm	4.32 ± 3.81 ^a^	5.11 ± 4.75 ^a^

**Table 3 cimb-47-00457-t003:** Changes in gill enzyme activities of rainbow trout exposed to 1.8 ± 1.0 °C for 30 days in comparison to fish acclimated to 9.4 ± 0.1 °C. Data are the mean and standard deviation (n = 10) and those superscripted by different letters are significantly different (*p* < 0.05).

Enzyme Activities (µmol/min/g Protein)	9.4 ± 0.1 °C	1.8 ± 1.0 °C
Enzymes with significant differences (*p* ≤ 0.05)		
Lactate dehydrogenase	153.15 ± 42.09 ^a^	108.65 ± 40.70 ^b^
Pyruvate kinase	5.61 ± 2.39 ^a^	17.27 ± 3.69 ^b^
Malate dehydrogenase	7.13 ± 3.73 ^a^	15.15 ± 7.26 ^b^
Enzymes without differences (*p* > 0.05)		
Carbonic anhydrase	4.75 ± 1.84 ^a^	4.19 ± 3.89 ^a^
Caspase 3	0.95 ± 0.57 ^a^	1.11 ± 0.79 ^a^
Na^+^/K^+^ ATPase	1.47 ± 0.48 ^a^	4.20 ± 2.15 ^a^
H^+^ ATPase	1.05 ± 0.51 ^a^	1.11 ± 0.79 ^a^

## Data Availability

The data presented in this study are available in the Appendix A.

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
