# Peer review of "Morphometric and Enzymatic Changes in Gills of Rainbow Trout After Exposure to Suboptimal Low Temperature"

_cimb, 2025, doi:10.3390/cimb47060457_

Round 1

Reviewer 1 Report

Comments and Suggestions for Authors

  This manuscript investigates the effects of suboptimal low temperature conditions on gill morphology and enzymatic activity in rainbow trout. The study contributes to understanding fish acclimation to cold environments. While the study provides insightful information regarding the effects of suboptimal low temperature on fish, some issues need to be resolved and clarified. Please see my following comments:  

1.Abstract: It is recommended to clearly describe the experimental groups, such as defining what constitutes the control group.

2.Introduction: A brief explanation should be provided for selecting these specific enzymes and genes as experimental indicators.

3.Line 93: "Water temperature of the control and experimental stream channels is shown in Figure 1" – Please provide the correct Figure 1, as the current Figure 1 depicts gill morphology. 4.Materials and Methods: Correct the numbering of the 7 subsections (e.g., change "3.1. Fish" to "2.1. Fish" for sequential consistency).

5.Line 113: Reference 14 does not include gill analysis; please replace it if incorrect.

6.Sample size discrepancies: Line 114 states "It was performed on 7 individuals per treatment." Line 177 states "6 individuals per treatment." However, Line 100 mentions "10 fish per stream channel were haphazardly sampled." Clarification needed: Were specimens excluded? If so, please explain the rationale (e.g., mortality, outliers).

7.Discussion: It is recommended that the authors elaborate on the study's limitations and propose future research directions. This will enable readers to gain a more comprehensive understanding of the research field and inspire subsequent researchers to conduct further in-depth investigations.

Author Response

  1. Abstract: It is recommended to clearly describe the experimental groups, such as defining what constitutes the control group.

Response: Information was added: Please, see lines 12-13.

  1. Introduction: A brief explanation should be provided for selecting these specific enzymes and genes as experimental indicators.

Response: Information was added. Please, see lines 62-73.

  1. Line 93: "Water temperature of the control and experimental stream channels is shown in Figure 1" – Please provide the correct Figure 1, as the current Figure 1 depicts gill morphology.

Response: Correct Figure 1. was added. Please, see lines 100-115.

  1. Materials and Methods: Correct the numbering of the 7 subsections (e.g., change "3.1. Fish" to "2.1. Fish" for sequential consistency).

Response: Numbering was corrected to 2.1. Fish, …

  1. Line 113: Reference 14 does not include gill analysis; please replace it if incorrect.

Response: Reference was added:

  1. Lahnsteiner, F. Morphometric and Enzymatic Changes in Gills of Rainbow Trout after Exposure to Elevated Temperature—Indications for Gill Remodeling. Animals 2024, 14, 919. https://doi.org/10.3390/ani14060919
  2. Sample size discrepancies: Line 114 states "It was performed on 7 individuals per treatment." Line 177 states "6 individuals per treatment." However, Line 100 mentions "10 fish per stream channel were haphazardly sampled." Clarification needed: Were specimens excluded? If so, please explain the rationale (e.g., mortality, outliers).

Response: Sample size was clarified in “2.3. Sampling”. Please, see lines 104-107.

  1. Discussion: It is recommended that the authors elaborate on the study's limitations and propose future research directions. This will enable readers to gain a more comprehensive understanding of the research field and inspire subsequent researchers to conduct further in-depth investigations.

Response: Information was added. Please, see lines 346-353.

Reviewer 2 Report

Comments and Suggestions for Authors

The article is interesting and well-written; however, major revisions need to be addressed (see attached pdf file for more details).

Author Response

Reviewer 2 made several annotations in the pdf of the manuscript. These annotation concern typos, format, and stylistic changes.

Response: We followed the suggestions of the reviewer throughout manuscript and made the suggested changes. The changes are marked in yellow in the revised form of the manuscript.

Round 2

Reviewer 2 Report

Comments and Suggestions for Authors

All the suggestions and comments were addressed by the authors; however, some minor revisions need to be made before the final acceptance (see pdf file for more details).

Author Response

Reviewer 3 made several annotations in the pdf of the manuscript. These annotation concern typos, format, and stylistic changes.

Response: We followed the suggestions of the reviewer throughout manuscript and made the suggested changes. The changes are marked in yellow in the revised form of the manuscript.